# Induced Systemic Resistance in the *Bacillus* spp.—*Capsicum chinense* Jacq.—PepGMV Interaction, Elicited by Defense-Related Gene Expression

**DOI:** 10.3390/plants12112069

**Published:** 2023-05-23

**Authors:** Blancka Yesenia Samaniego-Gámez, Raúl Enrique Valle-Gough, René Garruña-Hernández, Arturo Reyes-Ramírez, Luis Latournerie-Moreno, José María Tun-Suárez, Hernán de Jesús Villanueva-Alonzo, Fidel Nuñez-Ramírez, Lourdes Cervantes Diaz, Samuel Uriel Samaniego-Gámez, Yereni Minero-García, Cecilia Hernandez-Zepeda, Oscar A. Moreno-Valenzuela

**Affiliations:** 1Institute of Agricultural Sciences, Autonomous University of Baja California, Delta Highway s/n Ejido Nuevo León, Mexicali P.O. Box 21705, Baja California, Mexico; 2CONACYT—National Technological Institute of Mexico, Technological Institute of Conkal, CONACYT, Tecnológico Ave. s/n, Conkal P.O. Box 97345, Yucatán, Mexico; 3National Technological Institute of Mexico, Conkal Institute of Technology, Division of Graduate Studies and Research, Av. Tecnológico s/n, Conkal P.O. Box 97345, Yucatán, Mexico; 4Regional Research Center “Dr. Hideyo Noguchi”, Cell Biology Laboratory, Autonomous University of Yucatan, Av. Itzáez, Nmbr. 490 by 59 St. Centro, Merida P.O. Box 97000, Yucatán, Mexico; 5Yucatan Center of Scientific Research, Plant Biochemistry and Molecular Biology Unit, 43 St., Nmbr. 130, Chuburna de Hidalgo, Merida P.O. Box 97200, Yucatán, Mexico; 6Yucatan Center of Scientific Research, Water Sciences Unit, 8 St., Nmbr. 39, SM 64, Mz. 29, Cancun P.O. Box 77500, Quintana Roo, Mexico

**Keywords:** *NPR1*, *PR10*, *COI1*, begomovirus, biological control, sustainable agriculture

## Abstract

Induced systemic resistance (ISR) is a mechanism involved in the plant defense response against pathogens. Certain members of the *Bacillus* genus are able to promote the ISR by maintaining a healthy photosynthetic apparatus, which prepares the plant for future stress situations. The goal of the present study was to analyze the effect of the inoculation of *Bacillus* on the expression of genes involved in plant responses to pathogens, as a part of the ISR, during the interaction of *Capsicum chinense* infected with PepGMV. The effects of the inoculation of the *Bacillus* strains in pepper plants infected with PepGMV were evaluated by observing the accumulation of viral DNA and the visible symptoms of pepper plants during a time-course experiment in greenhouse and in in vitro experiments. The relative expression of the defense genes *CcNPR1*, *CcPR10*, and *CcCOI1* were also evaluated. The results showed that the plants inoculated with *Bacillus subtilis* K47, *Bacillus cereus* K46, and *Bacillus* sp. M9 had a reduction in the PepGMV viral titer, and the symptoms in these plants were less severe compared to the plants infected with PepGMV and non-inoculated with *Bacillus*. Additionally, an increase in the transcript levels of *CcNPR1*, *CcPR10*, and *CcCOI1* was observed in plants inoculated with *Bacillus* strains. Our results suggest that the inoculation of *Bacillus* strains interferes with the viral replication, through the increase in the transcription of pathogenesis-related genes, which is reflected in a lowered plant symptomatology and an improved yield in the greenhouse, regardless of PepGMV infection status.

## 1. Introduction

The disease management caused by viruses represents high costs and generates serious economic losses in agricultural productions worldwide [1]. The begomovirus genera, family Geminiviridae, is the most diverse and widespread member of the family worldwide, and it includes more than 400 reported species, including *Pepper golden mosaic virus* (PepGMV) [2,3,4]. Since its appearance and further identification, this virus has been studied as it has caused severe economic losses, as well as because it is an excellent model for the elucidation of the plant–begomoviruses interactions [5,6,7]. PepGMV causes adverse effects in crops of the family Solanaceae, such as tobacco, tomato, and several species of pepper (*Capsicum* sp.) [6,7]. Of these species, *C. chinense* L. has been widely used as a model for food, biochemical, and phytopathological studies. Due to its economic importance, it has a short crop cycle that allows us to identify, detect, and know the distribution of viruses that affect solanaceous crops [8,9].

In Mexico, several studies have been performed in order to identify the mechanisms of infection and the interaction between vector (*Bemisia tabaci* Gennadius B biotype), host and PepGMV in order to propose strategies of management and control [8,10,11]. The management of viral pathosystems is based on the prophylaxis that prevents virus dispersion and on the plant tolerance in order to generate plants with resistance to viral infection through different pathways [12]. The main strategy for the virus control depends on the genetic resistance of the host, its interaction with the environment, and the efficiency of the synthetic pesticides in vector control [13]. However, in PepGMV infections, the high multiplication rates of *B. tabaci* have favored the development of resistant/tolerant populations in this vector to the application of chemical products [14,15].

In recent years, many studies have focused on generating new strategies and approaches during disease management, and currently, commercial products based on microorganisms, such as beneficial fungi and plant growth-promoting rhizobacteria (PGPR), are applied to agricultural crops [16,17,18,19]. *Bacillus* is a PGPR that promote growth, regulates the plant physiology, and confers protection against pathogens due to many of its properties, such as the biosynthesis of auxins (IAA), production of siderophores, fixation and solubilization of soil nutrients, and induced systemic resistance (ISR) via jasmonic acid (JA), salicylic acid (SA), and ethylene (ET) signaling pathways [17,19,20,21,22,23,24]. The ISR elicited by rhizobacteria is a mechanism that strengthens the plant defense system against attack from pathogens. However, the induction of the ISR as a sustainable choice for viral disease control has been little studied [25,26,27].

During the ISR, the genes and transcriptional factors of the SA and JA signaling pathways, dependent on the non-expressor of pathogenesis-related genes 1 (*NPR1*), participate and involve the synthesis of pathogenesis-related proteins [24,28,29]. JA and ET transduce extracellular stimuli recognized by cell receptors to a large number of target molecules, which affect a fully coordinated and highly specific intracellular response to external stimuli [30]. Some reports indicate that the Coronatine Insensitive 1 (*COI1*) gene regulates the JA signaling pathway that induces ISR [28,31]. Within the reported genes in these pathways, the pathogenesis-related protein family 10 (*PR10*) [32] is included. 

Recent studies have shown that the ISR improves photosynthesis and favors plants under biotic stress conditions [33,34]. Specifically, our previous research has shown that *Bacillus* strains enhance photosystem, and furthermore, promote ISR in *C. chinense* plants infected with PepGMV, showing an increased CO_2_ assimilation, a decrease in the transpiration rate, and increased water use efficiency, which caused less severe symptoms in PepGMV-infected plants, as well as an increase in yield and fruit quality [35,36]. In this sense, could *Bacillus* spp. increase the expression of the plant defense mechanisms at a genetic level? The objective of the present study was to analyze the effect of the inoculation of different strains of *Bacillus* on the expression of the genes involved in plant responses to pathogens during its interaction with *C. chinense* infected with PepGMV.

## 2. Results

### 2.1. The Severity of the Infection Caused by PepGMV

The severity of the infection caused by PepGMV in pepper plants cultivated in controlled conditions (growth chamber) was evaluated in a temporal course experiment at 0, 7, 14, and 21 dpi. The visible symptoms based on the severity scale (see Section 4.4 in Materials and Methods) appeared as the viral infection progressed; the symptomatology developed as expected (Figure 1A). In addition, it was possible to observe mean differences in the viral titer, which increased in plants with higher severity levels: 1 = 22.90 Relative DNA Accumulation (RDA); 2 = 1558.56 RDA; 3 = 1844.55 RDA; 4 = 2161.02 RDA (Figure 2B).

### 2.2. Induced Systemic Resistance by Bacillus spp. in Capsicum chinense Jacq. against PepGMV

Symptoms in plants infected with PepGMV were first observed at 7 days post-inoculation (dpi) (Figure 2B). The first symptoms were downward leaf curling and yellow spots. In plants treated with *B. subtilis* K47, *B. cereus* K46, and *Bacillus* spp. M9, the symptoms of yellow spots, leaf curl, and dwarfism, characteristic of a Level 1 PepGMV severity, were observed until 14 dpi (Figure 2H–J). Symptoms were attenuated in the treatment with the *B. subtilis* K47-PepGMV (Figure 2H). PepGMV-infected plants showed severe golden mosaics, leaf curling, and dwarfing symptoms (Figure 2G), which corresponded to Level 4 on the severity scale. Mock-inoculated plants did not present any symptoms (Figure 2A,F,K). 

Viral symptoms were less severe in plants infected with PepGMV at 21 dpi (Level 2 severity, Figure 2L) compared to 14 dpi, with Level 4 severity (Figure 2G). The symptoms in plants inoculated with the *Bacillus* strains were similar at 14 dpi with Level 1 severity (Figure 2M,O), except for the plants inoculated with the M9 strain, where yellow mosaics and leaf curling were observed, which corresponded to Level 2 severity (Figure 2O). PepGMV was detected in all the plants inoculated with the virus, and was determined by PCR amplification. Taken together, these results indicate that *Bacillus* spp. induces resistance to PepGMV in pepper plants.

### 2.3. Accumulation of PepGMV in Capsicum chinense

The relative accumulation of viral DNA was quantified using the PepGMV-AC2 gene as a qPCR target. Viral DNA was detected at 7 dpi in all treatments, with mean differences among them. In PepGMV-infected plants, the viral DNA titer was up to 25 times lower than the viral DNA detected in plants inoculated with the K47 strain, and 9 and 17 times lower than in those treated with the K46 and M9 strains, respectively (Figure 3). In contrast, a significant reduction in the viral accumulation was observed in plants inoculated with the *Bacillus* spp. K47, K46, and M9 at 14 and 21 dpi. The viral accumulation of PepGMV in plants without the inoculation of *Bacillus* spp. K47, K46, and M9 increased over time; the highest values were observed at 7 dpi and then decreased at 14 and 21 dpi, with mean differences among treatments (Figure 3). These results strongly suggest that the strains of *Bacillus* spp. are involved in the reduction in the viral titer in PepGMV-infected plants.

### 2.4. Effect of Plan Inoculation with Bacillus in Gene Expression of Genes Related with the ISR

After infection with PepGMV, the expression levels of the *CcNPR1* gene increased in pepper plants treated with *Bacillus*. A 3-fold increase was observed in leaves from inoculated plants with *B. cereus* K46 when compared to the mock treatment at 2 h post inoculaction (hpi). The expression levels of the *CcNPR1* gene were significantly higher in plants inoculated with *B. subtilis* K47 at 24 hpi (Figure 4A), with a 19-fold increase when compared to mock-treated plants. The expression levels of *CcNPR1* decreased as the PepGMV infection progressed in *Bacillus*-treated plants, although in PepGMV-inoculated plants without *Bacillus*, a 9.5-fold increase in *CcNPR1* was observed at 7 dpi when compared to the mock treatment (Figure 4B). The aforementioned results suggest that treatments with *B. cereus* K46 and *B. subtilis* K47 sharply increased *CcNPR1* gene expression in pepper plants infected with PepGMV, and the increase will depend on the strain used.

Gene expression levels of *CcPR10* gradually increased in plants treated with *B. subtilis* K47, which showed a 10-fold increase and 190-fold increase at 8 and 12 hpi, respectively, when compared to the mock treatments (Figure 4C). The *CcPR10* gene expression levels showed a 30-fold increase in plants treated with *Bacillus* spp. M9 at 21 dpi, when compared to mock treatments (Figure 4D). These results indicate an involvement of the *CcPR10* gene during the early stages of the PepGMV infection in plants inoculated with *B. subtilis* K47. 

*CcCOI1* gene expression increased significantly at 24 hpi in leaves treated with *B. subtilis* K47 and *Bacillus* spp. M9 when compared to mock-treated plants, with a 16- and 57-fold increase (Figure 4E), respectively. *CcCOI1* gene expression levels remained significantly higher in pepper plants treated with *Bacillus* spp. M9, when compared to mock treatments with a 16- and 27-fold increase at 24 hpi and 21 dpi, respectively (Figure 4F,G). The above results indicate that treatment with *Bacillus* spp. markedly increased the expression of the *CcCOI1* gene in pepper plants infected with PepGMV progressively at more advanced stages of the disease.

### 2.5. Greenhouse Yield of Plants during the Bacillus spp.–C. chinensee–PepGMV Interaction 

The agronomic traits evaluated at the greenhouse level were total yield, fruit number, and fruit weight, which were quantified at 200 dpi. Plants without *Bacillus* inoculation (H_2_O) and plants treated with the K47 strain had the highest yield across all analyzed treatments (Tukey, α = 0.05) (Table 1). The results showed that the yields obtained in plants treated with three strains of *Bacillus* and inoculated with the PepGMV had mean differences compared to the plants treated with water and the plants inoculated with the virus and without *Bacillus* spp.

The plants treated with the K47 and M9 strains produced a total number of fruits and had fruit weights statistically similar to control plants (H_2_O) (Table 1). All plants pretreated with *Bacillus* spp. strains produced fruits of higher weight than plants inoculated only with PepGMV (Table 1). Taken together, these results showed that seeds of *C. chinense* treated with different strains of *Bacillus* spp. increased the habanero pepper greenhouse production in plants infected with PepGMV (Table 1). 

## 3. Discussion

In this study, our results indicate that the accumulation of PepGMV DNA increased over time in plants without *Bacillus*. These results have been reported previously in symptomatic pepper leaves infected with PepGMV, which showed a high accumulation of viral DNA and RNA, due to high rates of replication and transcription [7]. In contrast, in this study, we observed that in plants treated with *Bacillus* spp., the PepGMV viral titer and the symptoms decreased over time. Different reports demonstrate that inoculation with *Bacillus* spp. can reduce viral replication in infected plants [26,37]. Viral replication and movement are fundamental processes in the cycle of the disease; if both are affected, the result is a low viral concentration, and as a consequence, a decrease in symptoms [38]. Given that the PepGMV infection is associated with mechanical wounding, a mock treatment was prepared; during the early phases of the experiments, increases in the *CcNPR1, CcPR10*, and *CcCOI1* was observed when compared to plants without mechanical damage. The observed up-regulations of these genes suggest that they are responsive, to some extent, to mechanical damage; this type of damage is caused by insect vectors (*B. tabaci*) during viral transmission [10,15]. 

Some studies indicate that *NPR1* is a plant gene, and it has been observed that its expression is at low levels in healthy plants [32]. Moreover, the evidence demonstrates that *NPR1* is a fundamental component of the pathway of the SA-mediated signal transduction pathway, inducing defense genes [16,39]. We observed that in plants inoculated with *B. cereus* K46, *CcNPR1* levels sharply increase at 2 hpi, and these increases are statistically higher than in plants infected with PepGMV and not inoculated with *Bacillus* strains. It was suggested that during ISR by *Bacillus* spp. in *C. chinense*, *CcNPR1* is involved in the defense response to PepGMV immediately after infection, and their behavior is specific to each *Bacillus* strain. Although most strains of *B. cereus* are recognized as pathogenic microbes for humans, there are some strains used in the bio-fertilization and biological control of plant viruses [40,41,42]. In contrast, in plants infected with PepGMV and without *Bacillus* inoculation, the highest expression of *CcNPR1* was observed at 7 dpi. Similar studies report that in *C. annuum* plants infected with *Euphorbia mosaic virus-Yucatan Peninsula* (EuMV-YP), the expression of *NPR1* increased at 7 dpi [32]. After the accumulation of SA in response to a pathogen attack, *NPR1* oligomer disassociates in the cytoplasm, and after this, the monomer is translocated into the nucleus, and together with TGA transcription factors, they induce the pathogenesis-related gene (*PR’s*) expression [38,39].

The PR protein family has a complex pattern of expression, and many members of this family of genes are differentially expressed under conditions of environmental stress and in response to pathogen attacks within the signaling pathway systemic acquired resistance (SAR) [24,39]. In this study, the transcript levels of *CcPR10* were highest at 8 hpi in plants inoculated with *B. subtilis* K47, and its expression was statistically higher than in plants without rhizobacteria. In this way, several studies have shown that the *Tobacco mosaic virus* (TMV), *Cucumber mosaic virus* (CMV), *Tobacco etch virus* (TEV), and *Tobacco vein mottling virus* (TVMV) trigger the activation of *PR10*, and this protein functions as a ribonuclease [43,44]. Furthermore, it was demonstrated that the inoculation of leaves in *C. annuum* L. ‘Bukwang’ with *B. amyloliquefaciens* 5B6 caused an increased in the expression of *PR10* during CMV infection [45]. Therefore, *PR10* could not only be activated during the SAR, but it also participates in the ISR in viral diseases. 

It has been known that the *COI1* gene is the central regulatory component of the signaling pathway of SA/JA, and it is required for the defense responses of plants [28,39]. Recently, studies have reported that the C2 protein of geminivirus suppresses the defense response signaling pathway mediated by jasmonates because this protein affects the functioning of complex SCFCOI1 [46,47]. However, our data indicate that the *CcCOI1* levels of *C. chinense* plants treated with *Bacillus* spp. increased consistently for 7 dpi during the disease caused by PepGMV. Similarly, in tobacco plants infected with TMV and inoculated with *Bacillus* spp., the activation of the genes *NtPR1*, *NtCOI1*, and *NtNPR1* has been observed, generating a modulated ISR by *Bacillus* spp. [28]. These results suggest that despite the disease caused by PepGMV, *Bacillus* spp. promotes ISR through increased levels of transcripts of *CcCOI1*. 

Multiple investigations on plant–rhizobacteria–pathogen interactions have demonstrated the benefits in disease resistance and increased crop yield [48,49]. In this study, we consistently observed that plants treated with *B. subtilis* K47 had less severity of symptoms and better fruit quality (larger fruits) despite viral infection than healthy plants. In this sense, in tomato plants treated with the *B. amyloliquefaciens* strain MBI600, resistance to *Tomato spotted wilt virus* and *Potato virus Y* was increased through the salicylic acid pathway [1]. Similarly, the application of *B. velezensis* CE 100 on strawberry plants controls fungal diseases and improves yield [50]. 

Previously, our studies showed that *B. subtilis* K47 increased the photosynthetic parameters of the plant and prepared it for stress situations, such as viral diseases. Furthermore, this study reveals that inoculation with *B. subtilis* K47 increases the defense gene expression of *CcNPR1*, *CcPR10,* and *CcCOI1*, decreases the viral titer and severity of symptoms, and increases yield. It is evident that *Bacillus* activates ISR through a complex network of mechanisms, not only defending the plant from pathogen attacks, but also allow it to continue producing fruit. These results could be interesting in evaluating their effect as complex bacteria on the response to biotic and abiotic stresses during agricultural sustainable production.

## 4. Materials and Methods

### 4.1. Selection of Plant Material, Germination, and Growth Conditions

Seeds of *C. chinense*, accession H-224, were used [51]. The disinfection procedure consisted of the immersion of *C. chinense* seeds in a solution of commercial bleach (sodium hypoclorite, 2% *v*/*v*) for 15 min; the seeds were then rinsed three times with sterile distilled water and air-dried in absorbent paper. The seed germination was performed in trays with 200 cavities filled with sterile substrate (peat moss). The substrate was moistened up to field capacity with distilled sterile water. The trays were placed in a growth room (25 ± 2 °C, photoperiod of 16/8 light/dark), and they were watered every two days and leaf-fertilized weekly at a dose of 1 gL^−1^ (UltraFol, Biochem systems, Querétaro, México. After 18 days of germination, the seedlings were transferred into 500 mL Styrofoam cups, filled with sterile substrate, and maintained in the same controlled conditions [36].

### 4.2. Bacillus sp. Inoculation

The strains of *B. cereus* K46, *Bacillus* spp. M9 (a mixture of *B. subtilis* and *B. amyloliquefaciens*), and *B. subtilis* K47 were used [35,36]. All the *Bacillus* strains were gown in Luria-Bertoni Broth (37 °C, 24 h) under agitation. The cell density was adjusted to 1 × 10^8^ cells mL^−1^, 10 mL saline solution (0.8% *v*/*v*), and the inoculation of seeds was performed as described previously [35]. 

### 4.3. PepGMV Infection by Bioballistics

Seedlings obtained from *C. chinense* seeds inoculated with the previously described *Bacillus* strains were infected with PepGMV by bioballistics, when they had 3 to 4 true leaves. A total of 1 µm gold particles (BioRad, Hercules, CA, USA) was mixed with 5 µg of DNA from each hemidimer (A and B), as described by Carrillo-Tripp et al. [52]. The treatments consisted of: (1) H_2_O (control); (2) Mock (control); (3) PepGMV; (4) *B. subtilis* K47 + PepGMV; (5) *B. cereus* K46 + PepGMV; (6) *Bacillus* spp. M9 + PepGMV. The mock treatment consisted of bombarding the seedlings with 1 µm gold particles without viral DNA [52,53]. The treated plants were grown under controlled conditions (25 ± 2 °C, photoperiod of 16/8 light/dark) for 28 days post-inoculation (dpi) with PepGMV [36].

### 4.4. Severity Scale in C. chinense Plants under Controlled Conditions

The symptoms caused in *C. chinense* seedlings bioballistically infected with PepGMV were evaluated at 9 and 15 dpi in a growth room under controlled conditions, the scale of the severity of the symptoms was modified from Samaniego-Gámez et al. [36], with the following values: 1. Golden mosaics; 2. Golden mosaics and leave distortion; 3. Golden mosaics, leave distortion, and chlorosis; 4. Golden mosaics, leave distortion, chlorosis, and leaf curling. The severity analyses were assessed from ten observations per treatment [54].

### 4.5. Viral Titer Determination in C. chinense

Leaves of *C. chinense* from systemically infected plants with PepGMV were sampled (1 g), and the total DNA was isolated with CTAB, as described by Doyle and Doyle [55] without modifications. The detection reactions for the virus were performed within a thermalcycler (TC-412, Techne, Bibby Scientific Ltd., Chicago, IL, USA), using a 100 ng of total DNA and primers that targeted the AC2 gene (Table 2). The amplification conditions consisted of: 1 cycle at 94 °C (5 min), 35 cycles at 94 °C (30 s), 58 °C (30 s), and 72 °C (30 s), with an extension of 72 °C (10 min). The viral quantification was performed by quantitative PCR (qPCR) in a StepOne Real-Time PCR system (Applied Biosystems, Life Technologies, San Francisco, CA, USA), using SYBRGreen qPCR kit Platinum SuperMix-UDG (11733046, Invitrogen, Carlsbad, CA, USA). The real-time thermalcycler conditions were: 1 cycle at 94 °C (5 min), followed by 35 cycles at 94 °C (30 s), 58 °C (30 s), and 72 °C (30 s).

A gene normalization analysis was performed in order to determine the most stable housekeeping genes, such as *Actin*, *18S*, *Gliceraldehyde-3-phosphate-dehydrogensae*, *Ubiquitin*, and *β-Tubulin*, with the geNorm software (v.3.0, qBASE, Beavercreek, OH, USA) The normalization showed that *β-Tubulin* was the most stable gene among the samples, which agrees with previous reports in *Capsicum* [56,57]. The viral titter was determined with the 2^−ΔΔCt^ method [56,58], normalized with *β-tubulin* (Table 2), and expressed as RDA.

### 4.6. Relative Expression of CcNPR1, CcPR10, and CcCOI1 by Real-Time PCR

To determine the transcript levels of *CcNPR1, CcPR10*, and *CcCOI1* genes, leaf tissue was collected during the time course of the disease, 0 h before infecting with the virus, 2, 4, 8, 12, and 24 h after infection with PepGMV, and 7, 14, and 21 days after infection with PepGMV. Total RNA was isolated from leaves with the TRIZOL reagent, as described by Chomczynski and Sacchi [44]. The isolated RNA was treated with TURBO DNase (2238, Ambion, Life Technologies, Sunnyvale, CA, USA), and 2.5 µg was used for the cDNA synthesis with the SuperScript III Reverse Transcriptase (18080-044, Invitrogen, Carlsbad, CA, USA) and oligodT_18_ (18418-012, Invitrogen, Carlsbad, CA, USA). The transcript quantification was performed in a StepOne Applied Biosystems Real-Time PCR system (Applied Biosystems, Life Technologies, Carlsbad, CA, USA), using SYBRGreen qPCR kit Platinum SuperMix-UDG (11733046, Invitrogen, Carlsbad, CA, USA) with 100 ng of total cDNA. Transcript quantification conditions for *CcNPR1*: 1 cycle of 94 °C (5 min), 35 cycles of 95 °C (30 s), 55 °C (30 s), and 72 °C (30 s). Transcript quantification conditions for *CcPR10*: 1 cycle of 94 °C (5 min), 35 cycles of 95 °C (30 s), 58 °C (30 s), and 72 °C (30 s). Transcript quantification conditions for *CcCOI1*: 1 cycle of 94 °C (5 min), 35 cycles of 95 °C (30 s), 58 °C (30 s), 72 °C (30 s). The results were normalized with *β-tubulin* (Table 2) and expressed as relative transcript levels with the 2^−ΔΔCt^ method [56,57,58].

### 4.7. Evaluation of Agronomic Parameters

Plants at 28 dpi were transferred into black polyethylene bags (400 gauge) of 5 kg capacity (35 cm diameter, 40 cm height) filled with sterile substrate, and maintained in a greenhouse under controlled conditions (30 ± 2 °C, 65 ± 3% HR and 1100 mmol luminous intensity). The agronomic parameters were performed as previously described [36].

### 4.8. Experimental Design

A complete randomized design was used with 30 plants per treatment. For qPCR experiments, three biological replicates per treatment were analyzed. PCR products were cloned, sequenced, and compared via nucleotide BLAST in the NCBI. The detection of mean differences in each treatment was examined by an ANOVA with the Tukey’s HSD test at *p* ≤ 0.05 (Statistica, v.7.0.0, StatSoft Hamburg, Germany).

## 5. Conclusions

The seed inoculation with *Bacillus* promotes the ISR in *C. chinense* plants, which is reflected in the reduction in the viral accumulation and the symptom severity, which suggests that *Bacillus* could participate in the ISR through various mechanisms, such as the inhibition of the viral replication and an increase in the transcription rate of defense genes. Therefore, the ISR is a mechanism that could be implemented in disease management programs in sustainable agriculture. Future research should be focused on the study of the viral movement in *Bacillus*-treated plants and the transcriptional profile of the *Bacillus*–plant–geminivirus interaction in order to elucidate the mechanisms involved at transcript and the protein profile of this type of interaction.

## Figures and Tables

**Figure 1 plants-12-02069-f001:**
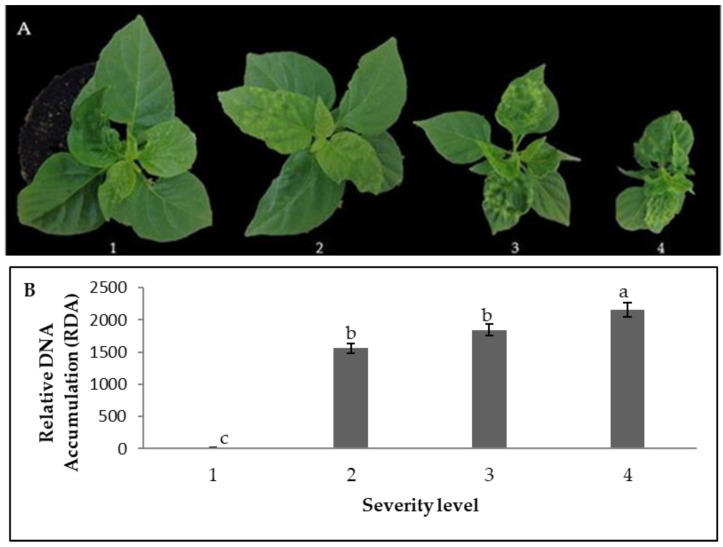
Symptom severity scale observed during infection caused by PepGMV in *Capsicum chinense* Jacq. (**A**) Pepper plants showing symptoms associated with each severity level from a scale from 1 to 4, and (**B**) relative DNA-PepGMV accumulation detected in pepper plants from each severity level of the scale. Different letters represent mean differences (Tukey, α = 0.05). Error bars represent the mean ± standard error, N = 3 with 3 technical replicates.

**Figure 2 plants-12-02069-f002:**
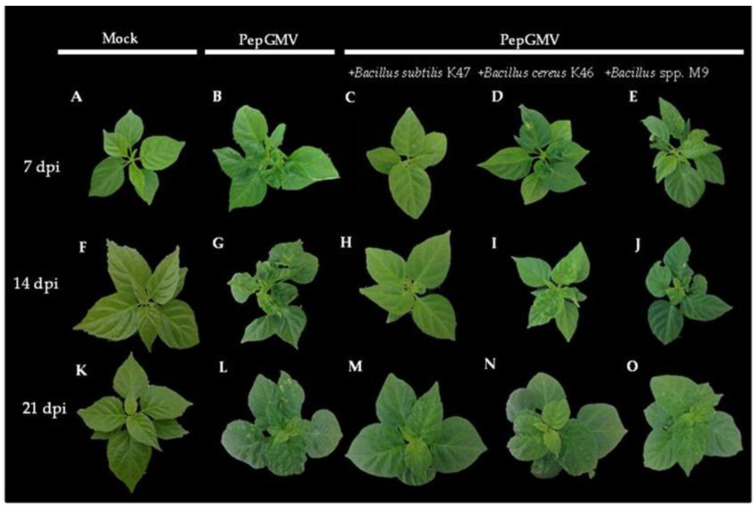
Induced systemic resistance by *Bacillus* spp. in plants of *C. chinense* infected with PepGMV. (**A**) Mock at 7 dpi; (**B**) Plant infected with PepGMV at 7 dpi; (**C**) *B. subtilis* K47-inoculated plant at 7 dpi infected with PepGMV; (**D**) *B. cereus* K46-inoculated plant at 7 dpi infected with PepGMV; (**E**) *Bacillus* spp. M9-inoculated plant at 7 dpi infected with PepGMV; (**F**) Mock at 14 dpi; (**G**) Plant infected with PepGMV at 14 dpi; (**H**) *B. subtilis* K47-inoculated plant at 14 dpi infected with PepGMV; (**I**) *B. cereus* K46-inoculated plant at 14 dpi infected with PepGMV; (**J**) *Bacillus* spp. M9-inoculated plant at 14 dpi infected with PepGMV; (**K**) Mock at 21 dpi; (**L**) Plant infected with PepGMV at 21 dpi; (**M**) *B. subtilis* K47-inoculated plant at 21 dpi infected with PepGMV; (**N**) *B. cereus* K46-inoculated at 21 dpi infected with PepGMV; (**O**) *Bacillus* spp. M9-inoculated plant at 21 dpi infected with PepGMV.

**Figure 3 plants-12-02069-f003:**
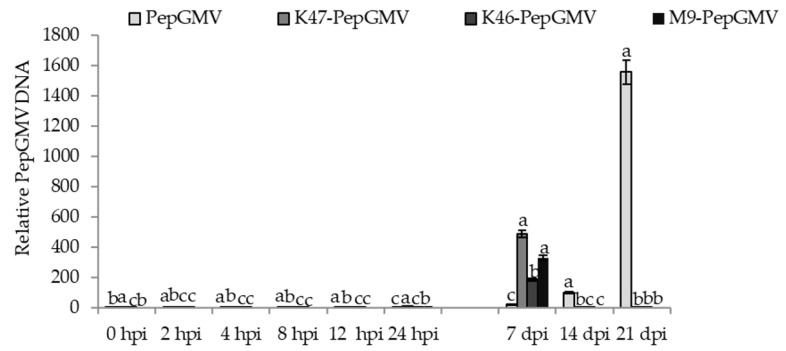
Viral DNA accumulation in pepper leaves inoculated with M9-PepGMV, K47-PepGMV, K46-PepGMV, and PepGMV. Different letters represent mean differences (Tukey, α = 0.05). Error bars represent the mean ± standard error; N = 3 with 3 technical replicates.

**Figure 4 plants-12-02069-f004:**
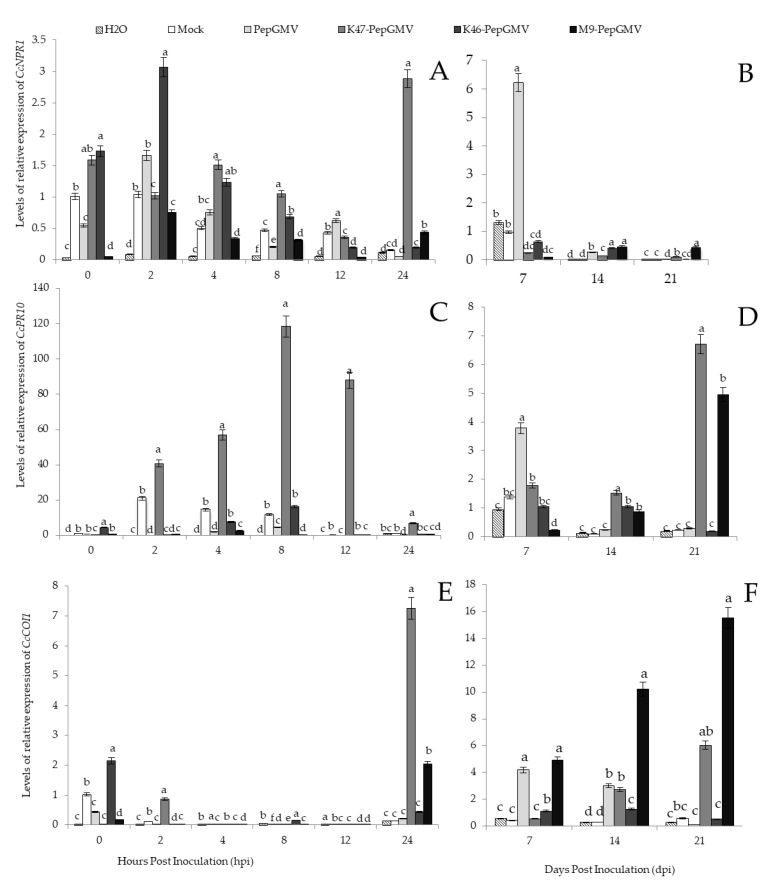
Levels of transcripts of *CcNPR1* (**A**,**B**), *CcPR10* (**C**,**D**) and *CcCOI1* (**E**,**F**) in *Capsicum chinense* Jacq plants inoculated with PepGMV, *B. subtilis* K47-PepGMV, *B. cereus* K46-PepGMV, *Bacillus* spp. M9-PepGMV. Different letters represent mean differences (Tukey, α = 0.05). Error bars represent the mean ± standard error; N = 3 with 3 technical replicates.

**Table 1 plants-12-02069-t001:** Average fruit weight (g), yield (g), and number of fruits in three harvests of habanero pepper plants inoculated with different bacterial strains, *B. subtilis* K47, *B. cereus* K46, *Bacillus* spp. M9, and infected with PepGMV. H_2_O = plants without viral infection and bacterial inoculation. Different letters represent mean differences (Tukey-HSD, α = 0.05).

Treatment	Mean Fruit Weight (g)	Fruits per Plant	Yield per Plant (g)
H_2_O	6.1 ± 0.05 b	52 ± 1.8 c	320.75 ± 6.51 c
PepGMV	5.3 ± 0.06 b	52 ± 1.4 d	274 ± 4.263 d
K47-PepGMV	7.9 ± 0.05 a	57 ± 1.5 a	451.75 ± 9.47 a
K46-PepGMV	7.6 ± 0.04 c	47 ± 1.3 b	358.75 ± 8.2 b
M9-PepGMV	7.7 ± 0.04 d	41 ± 1.1 b	314 ± 9.2 c,d

**Table 2 plants-12-02069-t002:** Primers used for real-time PCR.

Gene Name	Primer Sequences (5′-3′)	Fragment (bp)
*PepGMV AC2* [36]	GCCTTGTGGAGAGCTAATGC	213
	TTAGCGCAGTTGATGTGGAG	
β-*tubulin* [56]	TGTCCATCTGCTCTCTGTTG	204
	CACCCCAAGCACAATAAGAC	
*CcNPR1*	GAGGTGAGTTATGATGCTCTGG	141
	AACCAAGAAAGCCACTGCTG	
*CcPR10*	GCAGATGGAGGATGTGTTGG	147
	AGAAGGATTGGTGAGGAGGTAG	
*CcCOI1*	TGAAGAAGGTGCGGTTACAC	153
	ACCAGCCGAAAATCAGACAG	

## Data Availability

Not applicable.

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
