# Peer review of "Induced Systemic Resistance in the Bacillus spp.—Capsicum chinense Jacq.—PepGMV Interaction, Elicited by Defense-Related Gene Expression"

_plants, 2023, doi:10.3390/plants12112069_

Round 1

Reviewer 1 Report

Biological control of plant pathogens gains more and more attention of the scientists and farmers, nowadays. In Europe, the problem of a negative effect of chemicals used in agriculture on the environment and humans led to the EU strategy “Farm to Fork” within European Green Deal. In such context, the reported research on the possible use of Bacillus spp as elicitors of Induced Systemic Resistance against PepGMV has a great utility value in my oppinion.

I would like to emphasize, that the paper is very well written. However, some sentences should be corected due to editorial mistakes. English should be also improved thruought the manuscript.

In my opinion, the manuscript deserve to be published in Plants, an MDPI journal.

My general comments are as follows:

1.     Line 2, title – “spp.” should has dot at the end

2.     Lines 146, 193 – Species name should not contain “Jacq.”

3.     Line 220 – Bacillus should be written in italics

4.     Lines 280, 298, 307 – the sentence should not start from abbreviation

5.     Line 292 – text in parentheses should be deleted.

6.     Line 293 – should be only abbreviated name of the species. The full one was used in the text above.

7.     Line 295 – The sign “×” should be used

8.     Line 305 – “inoculation” instead of “infection” should be used

9.     Figure 5 – in my opinion, it is not readable due to high differences between yield and other two features (especially number of fruits). Table would be more suitable or three  smaller graphs.

10.  English should be improved, eg. line 286 (are instead of were).

Author Response

Response to Reviewer 1 Comments

Point 1: Line 2, title – “spp.” should has dot at the end

Response 1: The changes were made as the reviewer requested.

Point 2: Lines 146, 193 – Species name should not contain “Jacq.”

Response 2: The changes were made as the reviewer requested.

Point 3: Line 220 – Bacillus should be written in italics

Response 3: The changes were made as the reviewer requested.

Point 4: Lines 280, 298, 307 – the sentence should not start from abbreviation

Response 4: The changes were made as the reviewer requested.

Point 5: Line 292 – text in parentheses should be deleted.

Response 5: The changes were made as the reviewer requested.

Point 6: Line 293 – should be only abbreviated name of the species. The full one was used in the text above.

Response 6: The changes were made as the reviewer requested.

Point 7: Line 295 – The sign “×” should be used

Response 7: The changes were made as the reviewer requested.

Point 8: Line 305 – “inoculation” instead of “infection” should be used

Response 8: The changes were made as the reviewer requested.

Point 9: Figure 5 – in my opinion, it is not readable due to high differences between yield and other two features (especially number of fruits). Table would be more suitable or three  smaller graphs.

Response 8: Following your suggestion, the data are going to be displayed in the form of a table, in order to be readable.

Point 10: English should be improved, eg. line 286 (are instead of were).

Response 10: The changes were made as the reviewer requested.

Reviewer 2 Report

This manuscript showed the effectiveness of infection with Bacillus spp. on the defense against the DNA virus (PepGMV) in pepper plants. Although the results included several mistakes as pointed out later, the data themselves may be useful for many readers. However, there are several critical points that should be addressed before publication.

Major points

(1) Figures 1, 3, 4 and 5. There is no description about the biological replicate numbers. In addition, I cannot find the meaning of error bars; SD or SE?

(2) Line 149-150: viral DNA titer was lower in Bacillus-treated plants at 7 dpi. I cannot agree this description. In Figure 3, DNA titer of PepGMV sample was lowest among the all samples. It is clearly a mistake.

(3) line 167: The authors mentioned that levels of CcNPR1 mRNA in leaves of Bacillus-treated plants were higher than control sample. This will mislead the readers. Because viral infection is associated with mechanical wounding, it is easily expected that the expression of wounding-related genes are up-regulated. Therefore, the authors prepared the “mock” samples. So, the comparision between the mock samples and Bacillus-treated samples should be done and discussed. Similar comments should be applied to the description of PR10 and COI1 gene.

(4) Figure 4; I cannot find the positive effects of K46 when compared with the K47 and M9.

(5) Line 202, the authors should use “control” more clearly, because in Materials and Methods, the authors defined “control” is H2O treatment.

(6) Figure 5; I wonder if this figure’s data is replicated from the previsouly published data (Table 2 published in Pathogens 2021 10: 455). In addition, the numerical data are different. For example, mean fruit weights in Figure 5 of this manuscript are approximately 50 g, but the corresponding data in Table 2 of Pathogens (2021) are approximately 6-8 g. In contrast, the data of yield per plant are quite similar between Fig. 5 and Table 2. 

Minor points

(1) Abbreviations should be fully described at the first mention. Line 33, C. chinense should be Capsicum chinese. Line 105, what is RDA? Line 240, SAR should be fully described.

(2) Line 114, I wonder if Figure 1A may be Figure 2B.

(3) Line 132, there is no scale data for PepGMV samples.

(4) Line 251; sig-naling. Line 253; re-sponse

(5) Line 220; Bacillus should be italic style.

(6) Bacillus cereus is generally recognized as “pathogenic microbe for human”. The authors should prepare some description about the use of this bacteria on the risk as a food for human.

Author Response

Response to Reviewer 2 Comments

Major points

Point 1: Figures 1, 3, 4 and 5. There is no description about the biological replicate numbers. In addition, I cannot find the meaning of error bars; SD or SE?

Response 1: The changes were made as the reviewer requested. Added the information required by the reviewer in the figure captions.

Point 2: Line 149-150: viral DNA titer was lower in Bacillus-treated plants at 7 dpi. I cannot agree this description. In Figure 3, DNA titer of PepGMV sample was lowest among the all samples. It is clearly a mistake.

Response 2: The changes were made as the reviewer requested.

Point 3: line 167: The authors mentioned that levels of CcNPR1 mRNA in leaves of Bacillus-treated plants were higher than control sample. This will mislead the readers. Because viral infection is associated with mechanical wounding, it is easily expected that the expression of wounding-related genes are up-regulated. Therefore, the authors prepared the “mock” samples. So, the comparision between the mock samples and Bacillus-treated samples should be done and discussed. Similar comments should be applied to the description of PR10 and COI1 gene.

Response 3: The changes were made as the reviewer requested.

Point 4: Figure 4; I cannot find the positive effects of K46 when compared with the K47 and M9.

Response 4: The changes were made as the reviewer requested.

Point 5: Line 202, the authors should use “control” more clearly, because in Materials and Methods, the authors defined “control” is H2O treatment.

Response 5: The changes were made as the reviewer requested.

Point 6: Figure 5; I wonder if this figure’s data is replicated from the previsouly published data (Table 2 published in Pathogens 2021 10: 455). In addition, the numerical data are different. For example, mean fruit weights in Figure 5 of this manuscript are approximately 50 g, but the corresponding data in Table 2 of Pathogens (2021) are approximately 6-8 g. In contrast, the data of yield per plant are quite similar between Fig. 5 and Table 2.

Response 6: The changes were made as the reviewer requested. The data are similar because it is a replica of the procedure described in Pathogens 2021 10: 455. The data is presented in table format, the data are going to be displayed in the form of a table, in order to be readable.

Minor points

Point 1: Abbreviations should be fully described at the first mention. Line 33, C. chinense should be Capsicum chinese. Line 105, what is RDA? Line 240, SAR should be fully described.

Response 1: The changes were made as the reviewer requested.

Point 2: Line 114, I wonder if Figure 1A may be Figure 2B.

Response 2: The changes were made as the reviewer requested.

Point 3: Line 132, there is no scale data for PepGMV samples.

Response 3: The changes were made as the reviewer requested.

Point 4: Line 251; sig-naling. Line 253; re-sponse

Response 4: The changes were made as the reviewer requested.

Point 5: Line 220; Bacillus should be italic style.

Response 5: The changes were made as the reviewer requested.

Point 6: Bacillus cereus is generally recognized as “pathogenic microbe for human”. The authors should prepare some description about the use of this bacteria on the risk as a food for human.

Response 6: The changes were made as the reviewer requested.

Round 2

Reviewer 2 Report

This revised manuscript incorporated most my previous comments. However, some revision made it  more difficult to understand. Further revise is still necessary.

(1) Line 103, the sentence should be edited more correctly. scale cannot show the viral titer! In addition, the authors deleted the photograph showing symptoms, so it is difficult to understand the “visible scale”. My suggestion is as follows: The visible symptoms based severity scale (see Materials and Methods)…..

(2) Figure legends: N=3 with 3 technical replicate. Did this sentence mean that each datapoint was obtained from 3 biological samples and 3 technical replicates for each biological sample?

(3) Line 121-122: Viral symptoms were less severe in plants infected with PepGMV at 21 dpi compared to 14 dpi, with the level 3 of severity (Figure 2L?), while the symptoms in plants inoculated with Bacillus strains were similar than (than should be “to”?) the symptoms observed at 14 dpi with level 1 (?), except for the plants inoculated with the M9 strain in which were yellow mosaics and leaf curling were observed (level 2?). 

My comment is that please indicate the severity scale in each plant.

(4) Line 149: the authors should mention that the viral titers increased at 7 dpi and then decreased the following period (14 and 24 dpi). 

(5) Line 163-164: I cannot understand this sentence at all!

(6) Line 194: The authors indicated “mock”, but there is no data for mock in Table 1!

(7) Line 201: Still there is “mock”, see the comment (6). 

Author Response

Observations have been addressed, as requested by the reviewer.
